# Association between Interictal Epileptiform Discharges and Autistic Spectrum Disorder

**DOI:** 10.3390/brainsci9080185

**Published:** 2019-07-30

**Authors:** Laura Luz-Escamilla, José Antonio Morales-González

**Affiliations:** 1Laboratorio de Medicina de Conservación y Maestría en Ciencias de la Salud, Escuela Superior de Medicina, Instituto Politécnico Nacional, Plan de San Luis y Díaz Mirón S/N, Col. Casco de Santo Tomás, Alcaldía Miguel Hidalgo CP 11340, Mexico; 2Departamento de Higiene Mental, Hospital General Centro Médico Nacional “La Raza”, Instituto Mexicano del Seguro Social, Distrito Federal CP 02990, Mexico

**Keywords:** autistic spectrum disorder, EEG, interictal epileptiforme discharges, ADI-R

## Abstract

It has been reported that bioelectric alterations in an electroencephalogram (EEG) may play an etiological role in neurodevelopmental disorders. The clinical impact of interictal epileptiform discharges (IEDs) in association with autistic spectrum disorder (ASD) is unknown. The Autism Diagnostic Interview-Revised (ADI-R) is one of the gold standards for the diagnosis of autistic spectrum disorder. Some studies have indicated high comorbidity of IED and ASD, while other studies have not supported an association between the central symptoms of autism and IED. This review examines the high comorbidity and clinical impact of IED; patients with epilepsy are excluded from the scope of this review. ASD can be disabling and is diagnosed at an average age of 5 years old, at which point the greatest neurological development has occurred. If an association between IED and ASD is identified, a clinical tool that entails an innocuous procedure could enable diagnosis in the first years of life. However, in the absence of reports that prove an association between IED and ASD, patients should not be subjected to expensive treatments, such as the administration of anticonvulsant therapies.

## 1. Introduction 

Autism spectrum disorder (ASD) is a condition with different clinical manifestations that are expressed as a common clinical phenotype. The main characteristics of spectrum disorders include a deterioration in communication, a decrease in social interaction, and a restricted range of interests, as well as repetitive and stereotyped behaviors [1].

Interictal epileptiform discharges (IED) correspond to graph elements—generalized or focal paroxysms (acute waves, peak waves, or a complex of peak wave complexes)—which appear in an electroencephalogram (EEG) and are not accompanied by clinical manifestations [2,3].

The rate of interictal epileptiform discharges (IEDs) in patients with ASD has been typically reported to range from 6% to 30%, although a rate as high as 65% has also been reported [4,5,6,7]. Some children with autism may show DEI in their EEG that never manifests as seizures [8,9,10,11]. These data imply that early detection of IED would not affect the eventual development of epilepsy, but if we consider that the average prevalence of epilepsy in ASD doubles in adolescence [12], it is prudent to closely monitor ASD patients at this stage of their lives.

Current research on the anticonvulsant treatment of IED remains controversial. On the one hand, some studies have reported an improvement in affective instability, impulsivity, and aggression, but they have advised caution in the interpretation of their results [13]. On the other hand, other studies have not detected clinical improvement [14,15,16,17]. It should be noted that no study has reported a nootropic effect. This last point is important since a greater degree of cognitive impairment in ASD is an indicator of poor prognosis, as well as the impairment of linguistic abilities. Treatments that improve cognitive and linguistic skills in such cases are interventions such as stimulation, and they are the first line of treatment for ASD [18].

The measures to be taken should be carefully considered, given the inability to detect IED in a single EEG recording [19].

Furthermore, it is important to determine whether an association exists between IED and ASD: if so, we will have an element that would facilitate early diagnosis of ASD. If such an association does not exist, patients will not be subjected to anticonvulsant treatment.

## 2. Autistic Spectrum

Autism spectrum disorder (ASD) is a complex and heterogeneous disorder of neurological development. Autism itself is a syndrome, rather than an etiologically defined disorder, and it varies in its clinical presentation, ranging from high-functioning individuals to those who are severely affected [1]. Its current prevalence is 1 case per 68 children, with a male/female ratio of 4:1 [20,21]. A genetic basis and epigenetic and environmental factors are assumed to contribute to ASD [22,23]. Autism is primarily a syndrome of disconnection and is associated with aberrant connections between neural networks. The critical disconnection lies between the association cortex of the frontal lobe and the higher-order multimodal temporal lobe [24]. There are two peaks in the onset of epilepsy in ASD: One in early childhood and one in adolescence [5,25]. In a population-based study (*n* = 5815), the average prevalence of epilepsy in ASD doubled in adolescents [12].

The first ASD markers are present and detectable at the age of 12 months. The deficits related to ASD may be evident in the second or third year of the child’s life. At the age of 5–6 years, restrictive and repetitive behaviors increase. By adulthood, approximately 1–2% of people with autism are able to live independently, one-third have some degree of independence, and two-thirds require supervision [24].

Highly trained doctors can accurately diagnose ASD in children at the age of 2 years. Early behavioral signs include those that manifest as difficulties with attention, eye contact, smiling and exchanging affection, skills at play, and imitation [26]. Nevertheless, diagnosis is often delayed. A population-based study reported that the average age of diagnosis was approximately 5 years [26].

The delayed identification of ASD can be explained by several factors, including low diagnostic stability, limitations in the use of standardized instruments, the effect of the level of functioning on the symptoms of the smallest children, less stability in broader ranges of the spectrum (due to its heterogeneity), and the lack of qualified professionals to diagnose ASD [26]. Early diagnosis and intervention appear to be critical and are most effective in preschool-aged children [26]. The revised Autism Diagnostic Interview (ADI-R) has been characterized as a "gold standard" interview in the evaluation of children and adults with ASD [27].

Pharmacological treatment plays a secondary role compared with educational treatment (interventions such as stimulation). The purpose of medication is to treat the symptoms that interfere with the child’s integration in the home and school environment when behavioral techniques have failed. The approval of risperidone for the treatment of irritability in children aged 5–16 years who have autism has rendered it a "drug for autism" [28]. The lack of data on the prolonged use of risperidone and other drugs has necessitated thorough risk–benefit assessments [29]. In this way, treatment providers seek to minimize and prevent negative outcomes and to increasingly stimulate these children earlier and to a greater extent because, despite revealing a permanent problem, which is neither linear nor insurmountable, these children are receptive to treatment. Thus, the purpose of ASD treatment is to achieve a good level of social and mental functioning using various approaches, including educational or behavioral, psychotherapeutic, or psychopharmacological therapies [30,31,32,33].

## 3. Interictal Epileptiform Discharges

The presence of abnormalities in diagnostic tests such as the electroencephalogram (EEG), which measures the integrity and electrical stability of the brain, should be taken seriously. There is high comorbidity with neurodevelopmental disorders, but it is unknown whether their association is causal or an epiphenomenon. The answer to this question is of great importance since it would inform the clinical decision of whether to administer treatment.

Interictal epileptiform discharges (IEDs) are sharp waves or complex peak waves that occur in the absence of observable changes in behavior [34]. Spence et al. [2] defined them as points, sharp waves, slow waves, or complexes of generalized point waves, while Georges et al. [3] defined them as spikes, polypots, and sharp waves. IEDs correspond with generalized or focal pathological elements that appear in an EEG and are not accompanied by epileptic clinical manifestations. In some instances, IEDs are translated into fleeting neurological alterations that coincide with each other, have repercussions on everyday cognitive functions, and are observed at an early age, especially in academic performance and behavioral areas [2].

### 3.1. Neural Mechanisms that Are Probably Involved in Mediating the Harmful Effects of IED

IEDs arise from the hyperexcitability of neurons in the Central Nervous System (CNS). Neurons are depolarized in synchrony, and this generates an action potential. Both the origin of the initial event and the pattern of the propagation of the discharge determine the clinical aspect [35].

The cell membrane surrounding the soma rapidly depolarizes (100–200 ms) at high voltage (10–15 mV), and this depolarization generates a chain of action potentials that migrate from the soma through the axon. This unique and paroxysmal depolarization results in the surface EEG showing an interictal discharge. Subsequently, hyperpolarization continues, limiting the duration of the interictal paroxysm. This hyperpolarization is generated by a current transmitted through ion channels, including K^+^ channels that are activated by GABA and Ca^2+^ [35]. The slow wave that follows the spike temporarily alters the aspects of cortical function to a greater extent than the effect of the preceding spike. The observed effect corresponds to the anatomical location in which the events occur [36]. An IED inhibits ictal discharges in the entorhinal cortex of the hippocampus. There is sufficient evidence that discharges can have a negative effect on brain function through transient cognitive impairment (ICD), with direct effects on behavior and alteration of the physiological mechanisms involved in neuronal plasticity and memory [35]. The cognitive deterioration induced by IEDs is related to theta brain waves. It appears that theta brain waves play a crucial role in the mechanisms of synaptic plasticity, which is responsible for the formation and consolidation of memory, because long-term stimulation is facilitated by a theta-frequency stimulus [37]. IEDs can affect the consolidation of memory during sleep. During non-REM sleep, the decrease in cholinergic connections to the hippocampus results in the diffusion of excitatory activity in the CA3 circuits (the shape of the hippocampus in its entirety is that of a curved tube and reminds anatomists, a seahorse. At present, this part of the hippocampus is designated CA1, CA2, CA3, and CA; it is CA3 that facilitates connections between CA1 and the entorhinal cortex (EC) [38]. This results in high CA3 activity that propagates to CA1, depolarizes the pyramidal cells, and causes sharp waves. It is this phenomenon that seems to be involved in the transmission of feedback information to the EC and, subsequently, to the neocortex, allowing the consolidation of recently acquired information [39]. Molecular biology studies have shown the proteins required to maintain long-term potentiation are expressed in the hippocampus and the neocortex during sleep [39].

Edward Faught [40] reported that IED causes brain dysfunction. The proposed causal sequence is as follows: (1) A lesion occurs in the cortex; (2) transcriptional pathways are induced; (3) lateral recurrent connections increase; (4) neuronal hyperconnectivity occurs; and (5) the synchronous loading of several neurons manifests as an IED. The mechanism of this chain of events has been studied by analyzing the cortical tissue of people with epilepsy. This tissue is characterized by a prominent transcriptome that involves the activation of the mitogen-activated protein kinase (MAPK) pathway [41,42,43,44].

Another study aimed to explain how IED in children interferes with normal neuronal physiology and causes the interruption of various cognitive processes, such as plasticity, memory coding, and language processing [45]. These children deviate from the normal cognitive trajectory. The discharges can have a detrimental effect on intrinsic connectivity networks in the brain, reflecting a deficient organization of functional networks at rest and, in turn, abnormal neurocognitive development. The magnitude of the IED in the topology of the network, and resistance/vulnerability are associated with neurocognitive results. From functional magnetic resonance (FMR) imaging, the following main findings have been reported: (i) Large-scale changes in the network precede and follow IED; (ii) resistance to the topologies of the IED network is associated with a network at rest with the strongest connectivity; and (iii) vulnerability to IED is associated with poorer neurocognitive results [45]. Children with greater resistance to changes related to IED appear to have significantly stronger contributions from the posterior cingulate, the precuneus, the ventromedial prefrontal cortex, and the anterior cingulate. Those who are more vulnerable have significantly stronger contributions from the intraparietal circuit and the cortex of the anterior ventral cingulate, which implies a weaker network [45]. In particular, the thalamus can play an important role in the oscillatory measurement of the resistance of the network, as well as in the expression of the IED.

The impact of point-wave complexes, especially serial complexes, was shown to be stronger than that of isolated points. It has been proposed that the effect of the IED starts precisely before the point and ends with the termination of the slow-wave start [36,43].

According to reports, cognitive impairment is more common when the activity of point-wave discharges is generalized to 3 Hz with prolonged duration (more than 3 s) [44,46]. Shorter episodes of IED do not exert an evolutionary impact on cognitive function [47,48]. It seems that most reports on the cognitive effects during an IED involve episodes that last for approximately 3 s, but they can persist for longer than 3 s and reach the clinical threshold of crisis [49]. IED in the left cerebral hemisphere especially affects verbal functions, while those derived from the right hemisphere have repercussions on visuospatial functions [50].

A focal IED can alter the cognitive functions that correspond to each cortical region. In turn, the cognitive activity can suppress or activate discharges in a specific area. Simple motor tasks and the measurement of reaction time are less sensitive to these temporary impairments, while tasks involving memory and language are more sensitive [51]. EEG discharges generally occur more readily when the individual is in a relaxed state with their eyes closed, and the discharges are often absent during psychological tests [51]. Transient Cognitive Impairment (TCI) can be observed in approximately 50% of the patients who experience IEDs during tests [51]. With respect to the sensory afferent pathway, IEDs occur more frequently when individuals perform tasks that have a visual input mode [49]. Some studies have reported the long-term effect of IEDs on neurocognitive function. Brinciotti et al. [52] and Tuchman and Rapin [53] suggested that there may be a cumulative effect in patients with frequent episodes of IED.

Also reported correlation of diurnal IED load with specific functions, such as information processingspeed and short-term memory [34]. However, the effects of laterality have not always been observed [34]. IEDs that increase during sleep are related to a lower intelligence quotient (IQ). A greater reduction in IQ is associated with a greater appearance of IED during sleep. The effects of IEDs on sleep have been associated with low scores in language, reading, spelling, and numerical operations. Chronic exposure to IED, higher IED frequency, bilaterality, and increased sleep–wake discharges are associated with low cognitive performance [34].

The most common sites reported for IEDs include the right temporal lobe and the bilateral temporal, frontal, and occipital lobes, and parasagittal points have been observed in some patients, with the generalized point-wave complex found in 16% of the recorded cases. The most frequent site reported for IEDs is the frontal lobe [54]. Some studies have claimed that the most frequent location is the temporal lobe [8,55], but others have not supported this finding [9,56].

There are several authors who are regarded as pioneers in the study of IED. In 1936, Gibbs et al. [57] reported that there are no obvious clinical symptoms of epilepsy in patients who showed IEDs. These authors were the first to associate a transient alteration of the upper cortical functions with the clinical symptomatology of "masked epilepsy" or "larval epilepsy".

In 1939, Schwab [58] used a reaction test during the EEG and reported that the IED was related to deficiencies in cognitive function, the lack of response to stimuli presented during the generalized discharges of wave peaks, or extended reaction times. In 1984, Aarts et al. [46] were the first group to use video surveillance for such studies. These authors introduced the term Transient Cognitive Impairment (TCI). (see Table 1).

### 3.2. Interictal Epileptiform Discharges in Healthy Children

In asymptomatic individuals without prior signs of epilepsy, the prevalence of IED ranges from 0% to 6% in children [2,62,63,64]. The importance of these IEDs found in otherwise healthy individuals is not clear because the subsequent development of epilepsy is rare (approximately 6% in children) [65].

### 3.3. Epileptiform Discharges in Children with Autism Spectrum Disorder

There is little information that can be used to evaluate the association between interictal epileptiform discharges and autistic spectrum disorder. Most of the studies on this topic have concluded that there is a high percentage of comorbidity between IED and ASD [4,8,10,11]. Some children with autism may have IEDs in their EEG and never experience seizures [8,9,10,11].

Boutros cited the term “isolated epileptiform discharges” in reference to psychiatric patients who showed epileptic discharges but did not have epilepsy [19].

#### 3.3.1. Evidence in Support of the Association between Interictal Epileptiform Discharges with Autistic Spectrum Disorder

Most studies have reported frequency measurements that corroborate broad comorbidity. The reported rates are variable, ranging from 6.7% to 65% [4,5,6,7,9,54]. Many studies have reported rates of around 30% [55,66,67], and others have reported much lower rates [68,69]. In a study conducted by Akihiro et al. [54], the authors reported a comorbidity rate of 85.8% of IED and ASD (870/1014); this is the highest percentage reported. Similarly, these authors detected an inverse relationship between the IQ of the patient and the rate of appearance of points. These findings are important because they coincide with the data reported for the presence of only IED (without comorbidity with autism), which implies similar behavior. The finding on IQ is compatible with other reports that found a high correlation between the presence of IED and a low IQ [53,54]. However, other studies could not reproduce these results [11,66]. The decrease in IQ has been associated with the frequency of IED (≥1 discharge/10 s) and bilateral distribution [34].

In a study by Mulligan and Trauner [56], the authors reported that early epileptiform activity renders a patient particularly vulnerable to effects such as limited plasticity and inadequate neural networks, which in turn lead to the cognitive, language, and social deficiencies observed in autism, in addition to stereotyped behaviors. Their results suggested that the severity of autistic symptoms may be associated with a higher probability of epileptiform anomalies. At present, it is unknown whether treatment alters the result.

IED in a subgroup of autistic children pointed to more serious diseases, serious behavior problems, and severe social deterioration, as observed during a 2-year follow-up period [70].

Frontal abnormalities in EEGs that were mainly bilateral, as well as persistent hypsarrhythmia, were significantly related to the emergence of autistic behavior. The authors suggested that paroxysmal discharges in cortical areas undergoing rapid maturation may be involved in the development of autistic features [19].

Similarly, it has been indicated that the presence of frontal paroxysms has a significantly higher association with the subsequent development of epilepsy compared with centrotemporal paroxysms [71].

More than a decade ago, it was proposed that EEGs should be used as a selection procedure in ASD, but it was concluded that there was insufficient evidence to confirm or disprove this as a clinical guideline [72]. On the other hand, in a 2018 study, performing routine EEGs for children with ASD was suggested [73].

#### 3.3.2. Evidence against of the Association between Interictal Epileptiform Discharges and Autistic Spectrum Disorder

Autism may develop in the absence of detectable epileptic peaks. It is very probable that autism is a heterogeneous disorder, similar to all other psychiatric disorders. Therefore, the presence of epileptic discharges could represent an endophenotype that may help decrease the heterogeneity of the disorder [19].

Regression in neurological development is found to be present in ASD, and some studies have reported a high incidence of IED in autism, with clinical regression data of 33–64% [69,74], while other studies have found no such association between IED and regression [4,5].

Tetsu et al. [30] did not find a pathogenic relationship between IED and grade 1 ASD in children, suggesting an epiphenomenon or compensatory changes. This finding is very important since it does not associate IED with autism. Similarly, Milovanovic [75] noted that IEDs did not exert a significant effect on the core symptoms of autism according to assessments by the ADI-R.

The knowledge gained from the above studies is important for the guidance of clinical decisions. On the one hand, it is necessary to determine the type of discharges that are associated with a cognitive-behavioral deficit, which is associated with ASD. At the same time, clinicians must bear in mind that, at present, there are data that do support the existence of an association between IEDs and ASD or ADI-R domains [75], and these studies label it as an epiphenomenon [30].

## 4. To Treat or Not to Treat

### 4.1. Evidence in Favor of Treatment

In one study, 176 patients with epileptic discharges were treated with valproic acid. In follow-up EEGs (an average 10 months later), 46.6% of the patients had normal recordings, and an additional 17% showed improvement. None had worsened. The authors concluded that a more proactive approach to evaluation and treatment was justified because some of the abnormalities might be reversible [8]. This conclusion is supported by another study [76], in which the authors show that treatment of interictal epileptiform discharges can improve behavior in epileptic children with behavioral problems.

A study with divalproex sodium indicated that this treatment might be beneficial for patients with autism spectrum disorders, particularly those with associated characteristics such as affective instability, impulsivity, and aggression, as well as those with a history of EEG abnormalities. However, the authors indicated that these findings should be interpreted with caution, given the open retrospective nature of the study. Controlled trials are needed to replicate these preliminary findings [13].

Another study indicated that Divalproex sodium may be beneficial to patients with autism spectrum disorders, particularly those with the associated features of affective instability, impulsivity, and aggression, as well as those with a history of EEG abnormalities or seizures. It is noteworthy that all patients with an abnormal EEG and/or a history of seizures were rated as responders. However, these findings must be interpreted with caution, given the open retrospective nature of the study. Controlled trials are needed to replicate these preliminary findings [77].

Levetiracetam was reported to reduce hyperactivity, impulsivity, mood instability, and aggression in autistic children with these problems. No nootropic effect was observed [78]. The results of another study suggested that early etiological diagnosis and strict control of IEDs with anticonvulsants can prevent the worsening of cognitive function [79].

### 4.2. Evidence against Treatment

In a major randomized control trial by Hirota, et al. [14], anticonvulsants were found to be no better than the placebo.

A review by Tharp [17] concluded that there is no justification for the use of anticonvulsant medications or surgery in children with ASD without seizures.

Other studies have not found clinical improvement in patients after treatment with lamotrigine [15] or levetiracetam [16].

A large proportion of children with ASD and abnormal EEGs will never develop a seizure disorder [8,9,10,11].

### 4.3. Summary of the Evidence

Evidence for the effectiveness of anticonvulsants in reducing IED and autistic symptoms is based mainly on case studies. There have been no studies on this subject with adequate statistical design. Most of these studies have reported improvement in the behavior of autistic children, but none of them have reported a nootropic effect. 

Divalproex sodium has shown efficacy as a modulator of mood, aggressiveness, and impulsivity in other psychiatric conditions. Levetiracetam was reported to reduce hyperactivity, impulsivity, mood instability, and aggression in autistic children with these problems.

The cost-effectiveness of EEG treatment in children with symptoms of ASD has not been determined. 

The majority of these studies have not shown clinical value, the details of the studies are not specified. The majority of the negative studies began many years after the onset of symptoms, when the damage is already irreversible.

## 5. Future Research Directions and Conclusions

There is a significant gap in knowledge of whether IED and ASD are associated, and studies that answer this question are lacking.

Currently, it is unknown whether epileptiform interictal discharges and autism spectrum disorder are epiphenomena of the physiological process or if their association is derived from causality. There is no simple answer, and the distinction is not clear. Furthermore, the two possibilities are not mutually exclusive. There are numerous studies that support the negative effect of IED in other populations.

It is not known what role IED plays in the central symptoms of autism.

Few studies have analyzed the association between IEDs and ASD. The studies of Tetsu [30] and Milovanovic [75], which do not support an association between ASD and IED, are recent studies that present statistical significance. Few studies have reported clinical relevance [30,75].

On the other hand, it is unknown whether anticonvulsants actually improve symptoms and prognosis.

There have been no studies reported on autistic children without seizures or abnormal initial EEGs with follow-up EEGs as their disease develops (of course, unless a seizure occurs) [19].

The EEG is not indicated in the current clinical guidelines or the study protocol of ASD [72]; however, when considering the high comorbidity between IED and ASD, these recommendations should be reassessed [73] by conducting new guideline research. Also, given the inability to detect IED in a single EEG recording, the appropriate measures to take must be carefully considered. This represents a broad field for future research.

It is important to determine whether these two phenomena are associated: If they are associated, we need to develop potent pharmacological treatments to try to minimize damage to patients’ cognition and behavior.

On the other hand, if they are not associated, patients need not undergo pharmacological treatment with anticonvulsants, given the risk–benefit ratio.

It is important to treat children with autism spectrum disorder promptly and effectively since the first years of life are those in which the greatest neurological development occurs; thus, this stage represents the greatest opportunity to mitigate or prevent negative outcomes. Therefore, if the association exists, early intervention could have a strong impact on prognosis.

## Figures and Tables

**Table 1 brainsci-09-00185-t001:** Summarizes the first several studies on the effects of interictal epileptiform discharges on cognition.

Author	Contribution
**Gibbs et al., 1936** [57]	This author and collaborators cited that epileptiform discharges are clear clinical symptoms, such as automatisms, involuntary movements, or alterations in consciousness, and that one is also able to observe epileptiform discharges in the EEG without evident clinical symptoms. These authors were the first to suggest that such changes in the EEG could be associated with a transitory alteration in superior cortical functions. These authors designated these symptoms “masked epilepsy” or “larval epilepsy”.
**Schwab, 1939** [58]	These authors described a group of 14 patients with slowed reactivity during episodes associated with epileptiform discharges in the EEG.
**Rausch et al., 1978** [59]	These authors reported that intracranial EEGs generate more precise recordings of brain activity than non-invasive EEGs. These data provided the opportunity to study cognitive processes in greater detail.
**Aarts et al., 1984** [46]	In 1984, Aarts et al. introduced the concept of Transient Cognitive Impairment (TCI) by combining the use of EEG recordings and neurocognitive tests. In one of their studies, the authors included 46 patients who were studied for epilepsy and compared them with patients who had generalized and left and right focal discharges. They found the highest error rates (37.5%) in patients with generalized symmetrical discharges. Predominantly left discharges were associated with errors in verbal tasks, and right discharges were associated with errors in non-verbal tasks. In another publication, the authors found alterations in cognitive function that coincided with epileptiform discharges in the EEG in 23 of 1059 patients (2.2%) who were sent for a routine EEG. When the authors only included patients who had more than one episode of epileptiform discharge in a 5 min baseline EEG without apparent crisis, they observed that nearly 50% of the patients experienced cognitive alteration during the epileptiform discharges in the EEG. This group of authors proposed the descriptive term Transient Cognitive Impairment for episodes of epileptiform discharges in the EEG that are associated with cognitive alteration. In this manner, this concept applies to an episode of transient cognitive alteration without the association of any other external clinical sign.
**Binnie et al., 1987** [50]	These authors included 91 patients who were confirmed or suspected to have epilepsy. They compared right and left focal points and found that one-half of their patients presented deterioration in verbal tasks, non-verbal tasks, or both.
**Siebelink et al., 1988** [60]	In this study, which included 21 children with epilepsy or suspected to have epilepsy, these authors found transitory cognitive impairment due to epileptiform discharges.
**Aldenkamp et al. 2005** [43]	The authors conducted a prospective, open, and comparative clinical study to analyze the cognitive alterations that occurred during, immediately prior to, or immediately after IEDs of 3 s or more. Their study suggested that transient cognitive impairment is only present in generalized epileptiform discharges.
**Holmes and Lenck-Santini, 2006** [35]	The authors carried out a study in which they suggested that generalized and frequent epileptiform discharges can affect cognitive capacities by interfering with learning and memory and the consolidation of the latter during sleep.
**Kleen et al., 2013** [61]	These authors included 10 patients with refractory epilepsy who were analyzed by intracranial monitoring. The authors found that IEDs that localized in the hippocampus interrupted the recovery and maintenance of memory, but not coding.

EEG = ElectroEncephaloGram; TCI = Transient Cognitive Impairment; IED = Interictal Epileptiform Discharges.

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
