# Peer review of "Association between Interictal Epileptiform Discharges and Autistic Spectrum Disorder"

_brainsci, 2019, doi:10.3390/brainsci9080185_

Reviewer 1 Report

The topic of the relationship between interictal discharges and autism spectrum disorder is highly relevant in terms of why they are present and any disease-related implications they may have.  However, the review paper overall is not structured in a way to adequately address this question.  The paper’s title is about IEDs and ASD, but a large portion of the paper discusses neuroanatomical associations with ASD phenotypic features, clinical presentation, and use of the ADI-R in diagnosis.  It is as if the authors wrote two separate review papers.  I would suggest starting with section 4 and omitting the first sections, since that is the main theme of the review.  Then the authors can incorporate how IEDs in different areas may affect the clinical symptomatology of ASD. 

It is also not clear how IEDs and the ADI-R are related. 

There are multiple grammatical errors that should be addressed.

Author Response

Prof. Dr. Germán Barrionuevo,

Editor-in-Chief

Dr. Tal Kenet,

Guest Editor

Brain Sciences

Dear Drs. Barrionuevo and Kenet,

We are very pleased with the results of the evaluation conducted by the Scientific Committee of your renowned journal of our manuscript entitled “Association between Interictal Epileptiform Discharges and Autistic Spectrum Disorder”, and have carried out the task of responding to the well-targeted and greatly appreciated observations provided by the reviewers. Special Issue "Disrupted Functional Connectivity in Autism Spectrum Disorder

In this document, please find, in yellow, the changes kindly suggested by the Reviewers, which now appear as changes in the new version of the manuscript. Many thanks.

We follow here with the corrections and suggestions that we carried out on the article, taking into account the reviewer commentary:

Reviewer 1

The topic of the relationship between interictal discharges and autism spectrum disorder is highly relevant in terms of why they are present and any disease-related implications they may have.  However, the review paper overall is not structured in a way to adequately address this question.  The paper’s title is about IEDs and ASD, but a large portion of the paper discusses neuroanatomical associations with ASD phenotypic features, clinical presentation, and use of the ADI-R in diagnosis.  It is as if the authors wrote two separate review papers.  I would suggest starting with section 4 and omitting the first sections, since that is the main theme of the review.

Answer: We appreciate the comments of the reviewer in relation to the structuring of the article. It has been modified to make the paper clearer and more consistent with academic standards. The paper is organized as outlined below:

1. Introduction

2. Autistic spectrum

3. Interictal epileptiform discharges

3.1 Neural mechanisms that are probably involved in mediating the harmful effects of IED

3.2 Interictal epileptiform discharges in healthy children.

3.3 Epileptiform discharges in children with autism spectrum disorder

3.3.1 Evidence in support of the association between interictal epileptiform discharges and autistic spectrum disorder

3.3.2 Evidence against the association between interictal epileptiform discharges and autistic spectrum disorder

4. To Treat or Not to Treat

4.1 Evidence in favor of treatment

4.2 Evidence against treatment

4.3 Summary of the evidence

5. Future research directions and conclusions

Then the authors can incorporate how IEDs in different areas may affect the clinical symptomatology of ASD.

Answer: The article was restructured, and the neuronal mechanism that is probably involved in the mediation of the detrimental effects of IEDs was incorporated into section 3.1. page 4

In section 3.3, the possible association between IED and ASD is described. Page 8

It is also not clear how IEDs and the ADI-R are related.

Answer: In the new structure, this section was removed

There are multiple grammatical errors that should be addressed.

Answer: English editing of the manuscript has been completed, and the grammatical errors were corrected. Thank you for your feedback.

As a general point, we wish to mention that the observation on the editing of the article in English has been punctiliously attended to, and a new review of the language of the entire manuscript has been carried out in order for this to be adequate.

We hope that our responses to the reviewers' observations are satisfactory and are available at any time for doubts and concerns related with this new version of our manuscript.

Sincerely yours,

Laura Luz Escamilla, MD, Master

José Antonio Morales, MD, PhD.

Instituto Politécnico Nacional, México

Reviewer 2 Report

Description: The manuscript deals with a VERY important topic. That is the issue of the clinical significance of inter-ictal epileptic discharges or IEDs in autistic children, particularly those without clinical epilepsy. This is extremely important and seriously under-investigated. As the authors rightfully state autism may be detected by the 2-3rd year of life but epilepsy may not appear for few more years or may not at all. All the while IEDs are likely to be affecting brain function. The authors tried to marshal available evidence that IED activity may be causing damage if left unattended.  As such, the effort is quite commendable and important. Nonetheless, the approach taken by the authors leaves much to be desired and perhaps necessitate a total or near total re-write as explained below.

Critique: Major issues:

It is advisable that the introduction be much shorter and more precisely define the problems existing with our current knowledge. For example: Does early detection of IEDs make a difference in the eventual development of epilepsy. If so, would early treatment make a difference? Does treatment of IEDs affect core autistic symptoms? What is an adequate EEG work-up? Is one normal EEG sufficient to completely rule out presence of IEDs?

The authors rightfully allude to the controversial nature of the evidence. Hence, it would be very useful to have sections for “evidence for” and “evidence against” then a summary integration of the evidence. This structure should be repeated for the various issues addressed in the manuscript.

I think we should have a major section dedicated to the importance of the questions being asked. Another section should be devoted for the evidence available. A third section is for the possible neural mechanisms likely at play in mediating the deleterious effects of IEDs. A suggested fourth section should be for available evidence from treatment studies, particularly the issue of early (early screening and detection of IEDs) vs. treatment starting some years later? Finally, the authors should share their expertise in providing some guidance on how future studies can address the raised issues.

It is not clear if the authors are aiming at examining inter-ictal discharges (i.e, in epileptic patients), or Isolated epileptic Discharges in non-epileptic children. If they are probing the effects of inter-ictal activity in epileptic patients they need to tell the readers how they plan to disentangle effects of seizures and AEDs from the effects of epileptic discharges?

Minor issues:

Abstract: Line 8, the sentence starting with “On the other hand…” is too long and totally not clear. The abbreviation AED appears without explanation. There is no CLEAR message conveyed by the abstract.

Introduction: Most readers of this paper will have significant knowledge regarding the clinical syndrome of ASD. A very brief summary and adequate references should suffice. I suggest a single paragraph for the clinical description including rating scales and assessments methodology.

The section on IEDs: Should first make a case of why this is an important issue. Th8is section has many assertions (e.g., first sentence) without references. This section needs to be divided with clear subheadings into IEDs in otherwise healthy children, in epileptic non-ASD, Epileptic-ASD, and pure ASD.

The last sentence before table 1, seems to have a word missing? …..determine the clinical…(what??).

Towards the middle of page 13, the authors seem to already arrive at the conclusion that” there is sufficient evidence that the discharges can exert a negative effect…. If they feel this conclusion is tru, then it should be moved to the Conclusion section of the paper and some recommendations should be based on that.

I am not sure if Figure 1 is informative? The 65% in the middle not clear what it refers to??

There is some important literature that is missed. For example, Swatzyna RJ et al: EEG for Children with Autism Spectrum Disorder: Evidential Considerations for Routine Screening. Journal of Child and Adolescent Psychiatry, 2018 Sep 14. doi: 10.1007/s00787-018-1225-x. [Epub ahead of print]. PMID: 30218395, suggested routine EEG screening for ASD children. The authors should comment whether they think this is a good suggestion or premature? This is very important as current guidelines do not suggest screening EEGs.

In Boutros NN: Standard EEG: A research Roadmap for Neuropsychiatry. Springer, 2014, there is a chapter on ASD that summarizes all the areas of lack of knowledge and possible ways to address them. I suggest authors to at least take a look at the chapter.

All in all, the authors have collected most of the relevant literature, what is needed is much better organization. Finally, I also suggest including a native English speaking author.

Author Response

Prof. Dr. Germán Barrionuevo,

Editor-in-Chief

Dr. Tal Kenet,

Guest Editor

Brain Sciences

Dear Drs. Barrionuevo and Kenet,

We are very pleased with the results of the evaluation conducted by the Scientific Committee of your renowned journal of our manuscript entitled “Association between Interictal Epileptiform Discharges and Autistic Spectrum Disorder”, and have carried out the task of responding to the well-targeted and greatly appreciated observations provided by the reviewers. Special Issue "Disrupted Functional Connectivity in Autism Spectrum Disorder

In this document, please find, in yellow, the changes kindly suggested by the Reviewers, which now appear as changes in the new version of the manuscript. Many thanks.

We follow here with the corrections and suggestions that we carried out on the article, taking into account the reviewer commentary:

Reviewer 2

Description: The manuscript deals with a VERY important topic. That is the issue of the clinical significance of inter-ictal epileptic discharges or IEDs in autistic children, particularly those without clinical epilepsy. This is extremely important and seriously under-investigated. As the authors rightfully state autism may be detected by the 2-3rd year of life but epilepsy may not appear for few more years or may not at all. All the while IEDs are likely to be affecting brain function. The authors tried to marshal available evidence that IED activity may be causing damage if left unattended.  As such, the effort is quite commendable and important. Nonetheless, the approach taken by the authors leaves much to be desired and perhaps necessitate a total or near total re-write as explained below.

Answer: We appreciate the reviewer’s comments on our manuscrip

Critique: Major issues:

It is advisable that the introduction be much shorter and more precisely define the problems existing with our current knowledge. For example: Does early detection of IEDs make a difference in the eventual development of epilepsy. If so, would early treatment make a difference? Does treatment of IEDs affect core autistic symptoms? What is an adequate EEG work-up? Is one normal EEG sufficient to completely rule out presence of IEDs?.

Answer: The introduction was shortened. Each question was answered with respect to the title of the article

Autism spectrum disorder (ASD) is a condition with different clinical manifestations that are expressed as a common clinical phenotype. The main characteristics of spectrum disorders include a deterioration in communication, a decrease in social interaction, and a restricted range of interests, as well as repetitive and stereotyped behaviors [1].

Interictal epileptiform discharges (IED) correspond to graph elements—generalized or focal paroxysms (acute waves, peak waves, or a complex of peak wave complexes)—which appear in an electroencephalogram (EEG) and are not accompanied by clinical manifestations [2,3].

The rate of interictal epileptiform discharges (IEDs) in patients with ASD has been typically reported to range from 6% to 30%, although a rate as high as 65% has also been reported [4-7]. Some children with autism may show DEI  in their EEG that never manifests as seizures [8–11]. These data imply that early detection of IED would not affect the eventual development of epilepsy, but if we consider that the average prevalence of epilepsy in ASD doubles in adolescence [12], it is prudent to closely monitor ASD patients at this stage of their lives.

Current research on the anticonvulsant treatment of IED remains controversial. On the one hand, some studies have reported an improvement in affective instability, impulsivity, and aggression, but they have advised caution in the interpretation of their results [13]. On the other hand, other studies have not detected clinical improvement [14–17]. It should be noted that no study has reported a nootropic effect. This last point is important since a greater degree of cognitive impairment in ASD is an indicator of poor prognosis, as well as the impairment of linguistic abilities. Treatments that improve cognitive and linguistic skills in such cases are interventions such as stimulation, and they are the first line of treatment for ASD [18].

No study has reported improvement in central autistic symptoms following pharmacological treatment of IEDs [13–17]. It is important to recognize that patients are treated, not EEGs, and if clinical data are lacking, pharmacological treatment should not be administered. The measures to be taken should be carefully considered, given the inability to detect IED in a single EEG recording [19].

Furthermore, it is important to determine whether an association exists between IED and ASD: if so, we will have an element that would facilitate early diagnosis of ASD. If such an association does not exist, patients will not be subjected to expensive treatments.

The authors rightfully allude to the controversial nature of the evidence. Hence, it would be very useful to have sections for “evidence for” and “evidence against” then a summary integration of the evidence. This structure should be repeated for the various issues addressed in the manuscript.

Answer: The manuscript is structured in sections according to feedback and instructions.

3.3 Epileptiform discharges in children with autism spectrum disorder.

3.3.1 Evidence in favor of the association of interictal epileptiform discharges with autistic spectrum disorder

3.3.2 Evidence against the association of interictal epileptiform discharges with autistic spectrum disorder

4. To Treat or Not to Treat

4.1 Evidence in favor of treatment

4.2 Evidence against treatment

4.3 Summary integration of the evidence

I think we should have a major section dedicated to the importance of the questions being asked. Another section should be devoted for the evidence available. A third section is for the possible neural mechanisms likely at play in mediating the deleterious effects of IEDs. A suggested fourth section should be for available evidence from treatment studies, particularly the issue of early (early screening and detection of IEDs) vs. treatment starting some years later? Finally, the authors should share their expertise in providing some guidance on how future studies can address the raised issues.

Answer: The indicated sections were created.

2. Autistic spectrum (the importance of early detection and the impact on the prognosis is described)

3. Interictal epileptiform discharges

3.1 Neural mechanisms that are probably involved in mediating the harmful effects of IED

3.3 Epileptiform discharges in children with autism spectrum disorder

3.3.1 Evidence in favor of the association of interictal epileptiform discharges with autistic spectrum disorder

3.3.2 Evidence against the association of interictal epileptiform discharges with autistic spectrum disorder

4. To Treat or Not to Treat

4.1 Evidence in favor of treating

4.2- Evidence against treating

4.3 Summary integration of the evidence

5. Future research directions and conclusions

It is not clear if the authors are aiming at examining inter-ictal discharges (i.e, in epileptic patients), or Isolated epileptic Discharges in non-epileptic children. If they are probing the effects of inter-ictal activity in epileptic patients they need to tell the readers how they plan to disentangle effects of seizures and AEDs from the effects of epileptic discharges?

Answer: We intended to only examine the association between isolated epileptic discharges in children with autism. However, given the physiology, we returned to interictal epileptiform discharges.

Minor issues:

Abstract: Line 8, the sentence starting with “On the other hand…” is too long and totally not clear. The abbreviation AED appears without explanation. There is no CLEAR message conveyed by the abstract.

Answer: The correction was made, and the summary was modified. Page 1

Introduction: Most readers of this paper will have significant knowledge regarding the clinical syndrome of ASD. A very brief summary and adequate references should suffice. I suggest a single paragraph for the clinical description including rating scales and assessments methodology.

Answer: Taking this suggestion into account, we corrected the introduction tried to summarize it as much as possible.

The section on IEDs: Should first make a case of why this is an important issue. Th8is section has many assertions (e.g., first sentence) without references. This section needs to be divided with clear subheadings into IEDs in otherwise healthy children, in epileptic non-ASD, Epileptic-ASD, and pure ASD.

Answer: The referenced section begins by explaining why it is an important issue: "The presence of anomalies in diagnostic tests such as the electroencephalogram (EEG), which measures the integrity and electrical stability of the brain, should be taken seriously. There is high comorbidity with neurodevelopmental disorders, but it is unknown whether their association is causal or an epiphenomenon. The answer to this question is of great importance since it would inform the clinical decision of whether to administer treatment."

The references were adjusted, and the affirmations were eliminated.

Subtitles were created:

3. Interictal epileptiform discharges

3.1 Neural mechanisms that are probably involved in mediating the harmful effects of IED

3.2 Interictal epileptiform discharges in healthy children.

3.3 Epileptiform discharges in children with autism spectrum disorder. Boutros, cites the term of isolated epileptiform discharges, in psychiatric patients, referring to the presence of epileptic discharges in non-epileptic individuals. This article does not focus on patients with epilepsy

The last sentence before table 1, seems to have a word missing? …..determine the clinical…(what??).

Answer: Corrected. Page 6

Towards the middle of page 13, the authors seem to already arrive at the conclusion that” there is sufficient evidence that the discharges can exert a negative effect…. If they feel this conclusion is tru, then it should be moved to the Conclusion section of the paper and some recommendations should be based on that.

Answer: The article was restructured. The literature indicates that IED has a negative effect; however, when it is considered in combination with ASD, there is still a lot that is unknown.

I am not sure if Figure 1 is informative? The 65% in the middle not clear what it refers to??

Answer: It was deleted.

There is some important literature that is missed. For example, Swatzyna RJ et al: EEG for Children with Autism Spectrum Disorder: Evidential Considerations for Routine Screening. Journal of Child and Adolescent Psychiatry, 2018 Sep 14. doi: 10.1007/s00787-018-1225-x. [Epub ahead of print]. PMID: 30218395, suggested routine EEG screening for ASD children. The authors should comment whether they think this is a good suggestion or premature? This is very important as current guidelines do not suggest screening EEGs.

In Boutros NN: Standard EEG: A research Roadmap for Neuropsychiatry. Springer, 2014, there is a chapter on ASD that summarizes all the areas of lack of knowledge and possible ways to address them. I suggest authors to at least take a look at the chapter.

Answer: Added to the bibliographic references (specifically, 19 and 73), and the information was included.

All in all, the authors have collected most of the relevant literature, what is needed is much better organization. Finally, I also suggest including a native English speaking author

Answer: The article was restructured.

As a general point, we wish to mention that the observation on the editing of the article in English has been punctiliously attended to, and a new review of the language of the entire manuscript has been carried out in order for this to be adequate.

We hope that our responses to the reviewers' observations are satisfactory and are available at any time for doubts and concerns related with this new version of our manuscript.

Sincerely yours,

Laura Luz Escamilla, MD, Master

José Antonio Morales, MD, PhD.

Instituto Politécnico Nacional, México

Round  2

Reviewer 1 Report

The authors did a very nice job at all the extensive edits and feedback that were given. The manuscript is now much clearer.

Author Response

Reviewer 1

The authors did a very nice job at all the extensive edits and feedback that were given. The manuscript is now much clearer    

Answer: We appreciate the reviewer’s comments on our manuscrip

Reviewer 2 Report

This is a revised version of the manuscript entitled” Association between IEDs and ASD. I very much like the re-organization and I think the paper makes much more sense.

I only have one major issue with this version which is the evidence for clinical usefulness of AEDs regarding the central/core symptoms of ASD. They make an affirmative statement on page 2 lines 57 and 58 that “No study has reported improvement……”. Then they follow that with the VERY OLD statement that patients are treated and not EEGs. This statement simply obviates the need for their paper?? It is the crucial question of the paper and this line of research: to treat or not to treat.

They do not cite Hollander et al (2001) who conducted a retrospective pilot study to determine whether divalproex sodium was effective in treating core dimensions and associated features of autism.  They included 14 patients with either autism, Asperger’s syndrome or PPD-NOS. Subjects were included irrespective of h/o seizures or EEG abnormalities. Ten of the 14 patients who completed the trial (71%) were rated as responsive to treatment (mean dose 768mg/day and range of 125-2500 mg/day). Improvement was noted in core symptoms of autism as well as the associated features of affective instability, impulsivity, and aggression. Of note is that all patients with abnormal EEGs were rated as responders.

They then cite a study where 176 patients with ASD were treated with valproic acid and a substantial proportion showed EEG and even clinical improvement. They cite a book chapter by Boutros but not the original citation of the paper. They do need to give credit to the original publication. They do cite  Hollander et al, 2006 who conducted a Double-blind, Placebo-controlled study in 13 ASD patients and reported significant differences in repetitive behavior with a large ES (1.6).

These results contradict their “affirmative” statement in the Introduction.

Furthermore, there are serious issues with the “Evidence against treatment”: 1) Depriving patients with IEDs from treatment because medications are “expensive” is not an acceptable argument. AEDs are not more dangerous that antipsychotic agents that are routinely used.

2) Most studies that showed no clinical value have not given full detailed EEG accounts of the patients. Furthermore, these studies (not that I am aware off) have conducted EEGs upon first diagnosis of ASD and started Tx then. Most negative studies started tx many years after onset of symptoms and as the authors rightfully point out, the damage may be irreversible by then.

All in all, the manuscript is much improved but paying a little more attention to the above issues would further strengthen the paper.

Author Response

In this document, please find, in yellow, the changes kindly suggested by the Reviewers, which now appear as changes in the new version of the manuscript. Many thanks.

We follow here with the corrections and suggestions that we carried out on the article, taking into account the reviewer commentary:

Reviewer 2

This is a revised version of the manuscript entitled” Association between IEDs and ASD. I very much like the re-organization and I think the paper makes much more sense

I only have one major issue with this version which is the evidence for clinical usefulness of AEDs regarding the central/core symptoms of ASD. They make an affirmative statement on page 2 lines 57 and 58 that “No study has reported improvement……”. Then they follow that with the VERY OLD statement that patients are treated and not EEGs. This statement simply obviates the need for their paper?? It is the crucial question of the paper and this line of research: to treat or not to treat.

Answer 1st paragraph: Correction was taken into consideration. We removed the statement (page 2, lines 57 and 58), that "No study has reported improvements" and replaced it with: Current research on the anticonvulsant treatment of IED remains controversial. On the one hand, some studies have reported an improvement in affective instability, impulsivity, and aggression, but they have advised caution in the interpretation of their results [13]. On the other hand, other studies have not detected clinical improvement [14–17]. The original statement was deleted.

They do not cite Hollander et al (2001) who conducted a retrospective pilot study to determine whether divalproex sodium was effective in treating core dimensions and associated features of autism.  They included 14 patients with either autism, Asperger’s syndrome or PPD-NOS. Subjects were included irrespective of h/o seizures or EEG abnormalities. Ten of the 14 patients who completed the trial (71%) were rated as responsive to treatment (mean dose 768mg/day and range of 125-2500 mg/day). Improvement was noted in core symptoms of autism as well as the associated features of affective instability, impulsivity, and aggression. Of note is that all patients with abnormal EEGs were rated as responders.

Answer:  2nd paragraph: We quoted Hollander et al. 2001, reference 77, page 10. We eliminated the affirmation. Another study indicated that Divalproex sodium may be beneficial to patients with autism spectrum disorders, particularly those with the associated features of affective instability, impulsivity, and aggression, as well as those with a history of EEG abnormalities or seizures. It is noteworthy that all patients with an abnormal EEG and/or a history of seizures were rated as responders. However, these findings must be interpreted with caution, given the open retrospective nature of the study. Controlled trials are needed to replicate these preliminary findings [77].

They then cite a study where 176 patients with ASD were treated with valproic acid and a substantial proportion showed EEG and even clinical improvement. They cite a book chapter by Boutros but not the original citation of the paper. They do need to give credit to the original publication. They do cite  Hollander et al, 2006 who conducted a Double-blind, Placebo-controlled study in 13 ASD patients and reported significant differences in repetitive behavior with a large ES (1.6). These results contradict their “affirmative” statement in the Introduction.

Answer 3rd paragraph: The bibliographic citation in which the chapter of a book by Boutros was cited was corrected, the credit was granted to Chez, 2006, reference 8, page 9. We removed the statement. He added to Pressler, reference 76. This conclusion is supported by another study [76], in which the authors show that treatment of interictal epileptiform discharges can improve behavior in epileptic children with behavioral problems

Furthermore, there are serious issues with the “Evidence against treatment”: 1) Depriving patients with IEDs from treatment because medications are “expensive” is not an acceptable argument. AEDs are not more dangerous that antipsychotic agents that are routinely used.

Answer 4th paragraph: The argument for expensive medications was eliminated (it was an error in translation).

2) Most studies that showed no clinical value have not given full detailed EEG accounts of the patients. Furthermore, these studies (not that I am aware off) have conducted EEGs upon first diagnosis of ASD and started Tx then. Most negative studies started tx many years after onset of symptoms and as the authors rightfully point out, the damage may be irreversible by then.

Answer 5th paragraph. Correction was taken into consideration for the Summary of Evidence section. Evidence for the effectiveness of anticonvulsants in reducing IED and autistic symptoms is based mainly on case studies. There have been no studies on this subject with adequate statistical design. Most of these studies have reported improvement in the behavior of autistic children, but none of them have reported a nootropic effect. Divalproex sodium has shown efficacy as a modulator of mood, aggressiveness, and impulsivity in other psychiatric conditions. Levetiracetam was reported to reduce hyperactivity, impulsivity, mood instability, and aggression in autistic children with these problems. The cost-effectiveness of EEG treatment in children with symptoms of ASD has not been determined.  The majority of these studies have not shown clinical value, the details of the studies are not specified. The majority of the negative studies began many years after the onset of symptoms, when the damage is already irreversible.

All in all, the manuscript is much improved but paying a little more attention to the above issues would further strengthen the paper.

Answer: We appreciate the reviewer’s comments on our manuscrip

We hope that our responses to the reviewers' observations are satisfactory and are available at any time for doubts and concerns related with this new version of our manuscript.

Sincerely yours,

Laura Luz Escamilla, MD, Master

José Antonio Morales, MD, PhD.

Instituto Politécnico Nacional, México